# Thousand Cankers Disease in Walnut Trees in Europe: Current Status and Management

**DOI:** 10.3390/pathogens12020164

**Published:** 2023-01-19

**Authors:** Matteo Bracalini, Alessandra Benigno, Chiara Aglietti, Tiziana Panzavolta, Salvatore Moricca

**Affiliations:** Department of Agricultural, Food, Environmental and Forest Sciences and Technologies, University of Florence, Piazzale delle Cascine 28, 50144 Firenze, Italy

**Keywords:** *Geosmithia morbida*, *Pityophthorus juglandis*, invasive plant parasites, thermotolerant species, TCD containment, disease control

## Abstract

Thousand cankers disease (TCD) is a new deadly disease in walnut trees (*Juglans* spp.), which is plaguing commercial plantations, natural groves, and ornamental black walnut trees (*Juglans nigra*) in their native and invasion areas in the US and, more recently, in artificial plantations and amenity trees in the newly-invaded areas in Europe (Italy). This insect/fungus complex arises from the intense trophic activity of the bark beetle vector *Pityophthorus juglandis* in the phloem of *Juglans* spp. and the subsequent development of multiple *Geosmithia morbida* cankers around beetles’ entry/exit holes. After an analysis of the main biological and ecological traits of both members of this insect/fungus complex, this review explores the options available for TCD prevention and management. Special focus is given to those diagnostic tools developed for disease detection, surveillance, and monitoring, as well as to existing phytosanitary regulations, protocols, and measures that comply with TCD eradication and containment. Only integrated disease management can effectively curtail the pervasive spread of TCD, thus limiting the damage to natural ecosystems, plantations, and ornamental walnuts.

## 1. Introduction

*Juglans* species are highly valued for economic, social, and environmental reasons. Members of this genus are, in fact, valuable resources to humans for their nuts and high-quality wood, as well as ornamental landscape trees [1]. These species are also fundamental to wildlife as they constitute habitats for animals and provide fruits that serve as food for birds, squirrels, and other small mammals [2]. Among the walnut species, the black walnut *(Juglans nigra* L.) is one of the most appreciated in the US for its nuts, which have a sweet and distinctive taste, and for its dark brown and hard, strong wood. By virtue of these properties, the black walnut is considered one of the most valuable timber trees in the US, with an estimated value for standing black walnut of USD 568 billion [3]. The high appreciation for this species in Europe also led to a continued introduction of black walnut seedlings from the US starting from the end of the 17th century [1]. Because of this massive introduction campaign, the black walnut today is considered an acclimatized species in Western Europe [4]. However, the most widespread *Juglans* species in Europe is the English walnut (*Juglans regia* L.), a tree believed to be native to Asia, or possibly to Central Europe and the Balkans, which is considered to have been naturalized in the south-western countries of the continent [5]. *Juglans nigra* and *J. regia* are the two main species of walnut exploited in Europe for commercial wood and nut production, as well as for amenity trees; they also play an important environmental role.

Widespread mortality of black walnut trees was observed in Colorado, USA, starting from 2001 [6]. In the same year, the first report associating the walnut twig beetle (WTB), *Pityophthorus juglandis* Blackman (Coleoptera, Curculionidae, Scolytinae), with the widespread decline of black walnuts in New Mexico was published [7]. Subsequently, a fungus of the genus *Geosmithia* (Ascomycota, Hypocreales, and Bionectriaceae) was isolated from the phloem around WTB galleries in attacked walnuts [6]. These authors, therefore, established that walnut tree mortality was caused by an insect/fungal disease complex. This involves multiple WTB attacks followed by several cankers induced by the above *Geosmithia* fungus (Figure 1). For this reason, the authors named this formerly undescribed syndrome “thousand cankers disease” (TCD) [7]. In 2011, the mitosporic filamentous fungus *Geosmithia morbida* M. Kolařík, E. Freeland, C. Utley, and N. Tisserat (GM) was identified as the primary causative agent of cankers surrounding WTB galleries [8]. In the same year, Grant et al. [9] reported the first occurrence of TCD in the native range of black walnut in Tennessee. Finally, TCD was first reported in Europe, specifically in northern Italy (Veneto region), in 2013 [10]. Currently, the disease is widespread in several states of the US and in several regions of northern and central Italy (Table 1) [11,12,13].

While TCD has been a phytosanitary problem in various US states for at least 10 years, resulting in a high research output, the problem in Europe, to date, is limited to a single country. Therefore, deeper knowledge about the biology and ecology of both of the TCD biotic agents in Europe is necessary. This knowledge is key to refining quarantine regulations in Europe as well as designing monitoring guidelines for the two organisms for the specific conditions of the European territory, where black walnut grows only in residential, park, and plantation settings. Gaining more insights into the epidemiology of TCD is also crucial for training the personnel involved in phytosanitary surveys, the management of urban green areas, as well as in customs inspection services. This paper reviews the current status of TCD management in both the North American and European continents, reporting the state of knowledge of this harmful disease, describing its two causative agents, its main symptoms, its main tree hosts, other potential vectors, and the available control strategies. Particular emphasis is given to integrated pest management for TCD, highlighting some recent diagnostic developments and stressing the crucial aspects of surveillance, monitoring, eradication, containment, and public involvement.

## 2. The Pathogen *Geosmithia morbida*

*Geosmithia morbida* is a mitosporic ascomycete of the globally widespread genus *Geosmithia*, which includes species that produce nonviscous conidia on penicillate conidiophores [14]. The macro- and micromorphological features of GM have already been described by Kolařík et al. [8] and Moricca et al. [15]. GM colonies are slow growing, sublobate, planar, and produce a hyaline to whitish mycelium, with verticillate conidiophores producing metulae, phialides, and conidia [15]. GM has proven itself to be a thermotolerant fungus, with an optimal growth temperature of around 31 °C, although it is reported to survive up to 41 °C [8]. It also maintains its viability when inoculated on wheat seeds exposed to 48 °C [8]; however, Moricca et al. [16] observed a cessation of growth beyond 35 °C in the lab. 

Different species of *Geosmithia* are frequently associated with bark beetles that infest broad-leaved and coniferous trees [14], with variable degrees of specificity for both vectors and hosts. In fact, some of these fungus species are specialists associated with just a few insect vectors feeding on a single genus or family of plants; others are generalists associated with numerous vectors and hosts. With GM being restricted to the tree genus *Juglans*, it must be considered a specialist fungus [17]. However, some *Geosmithia* spp. are not associated with vector insects at all and live as endophytes within plants or as saprophytes in decaying wood, cereal residues, or in the soil [8,14]. GM is considered an apparently harmless wound pathogen, which needs an insect vector to enter the host tree [8]; in fact, its propagules are regularly transported by the WTB and are inoculated into walnut bark tissue during the digging of galleries [6,12]. This insect–fungus association was repeatedly confirmed by numerous observations, both in native area of TCD in North America [8,18,19] and in its introduction area in Europe [11,12].

*Geosmithia morbida* behaves as a weak pathogen in its native area but becomes a threat outside of its native range, causing necrosis in *J. nigra* phloem tissue and eventually leading the affected walnuts to death. This filamentous ascomycete is, together with *Geosmithia pallida* (G. Sm.) M. Kolarík, Kubátová, and Paotová, the agent of the foamy bark canker on *Quercus agrifolia* Nee, the only pathogenic species in the genus *Geosmithia* [19], with all the other congeneric species having evolved as symbiotic beetle-associated saprotrophs [17]. Shifts towards pathogenic behavior have occurred in different fungal lineages during their evolution and result from those adaptive changes that channel the trajectory of the symbiotic relationships towards parasitism [20]. When analyzing some adaptive traits of bark and ambrosia beetle-associated fungi, Veselská et al. [17] hypothesized that GM might have evolved virulence factors that may account for its switch from a saprotrophic to a pathogenic life strategy. GM is, in fact, the only member of the genus *Geosmithia* that produces a series of enzymes to degrade the structural components of the host cell wall cellulose, hemicellulose, and lignin, albeit with moderate enzyme production. Furthermore, GM possesses strong antibiotic capabilities that are necessary to defend itself from fungal mycoparasites [21].

## 3. The Vector *Pityophthorus juglandis*

*Pityophthorus juglandis* is a small bark beetle described in detail by Wood [22] and Seybold et al. [23]. The adults are 1.5 to 2 mm long and have a relatively narrow body (three times longer than wide); they are reddish-brown or brown. The forehead of the females, whose length does not exceed half the distance between the eyes, has a round brush of golden bristles; instead, the forehead of the males has very sparse setae, though sometimes a narrow brush of short bristles may be observed immediately above the mandibles. The pronotum is shield-shaped; it is located above and behind the head. The anterior half of the pronotum, seen frontally, tilts upwards, reaching its apex before the pronotum’s midpoint; it also presents four to six concentric arches of asperities (cuticular ridges). The elytra have close pits and short, sparse bristles. The apex of the elytra is rounded, and the hind declivity is shallow, often shiny. The downsloping of the elytra in females is smooth, while in the males, it has rows of tiny granules placed on the first and third interstrial rows [22,24]. Like most bark beetles, the larvae are white and cirtosomatic, with reddish-brown sclerotic heads, and live in the phloem [22].

The WTB has a long flight period, generally from spring to autumn, with more than one overlapping generation, whose number depends mainly on climatic conditions [25]. Its flight capacity is not very high; in fact, one-third of the individuals of the same population were estimated to fly less than 100 m [26]. Thus, the dispersal of the beetle is mainly sustained by human-mediated transport, as demonstrated in Italy, where a mean annual dispersal of 9.4 km was recorded [27]. The WTB preferentially attacks stressed trees; however, it can also exploit healthy ones [11]. It bores galleries in the phloem of its hosts, especially colonizing the lower portion of the branches that are less exposed to thermal excursions [28], particularly those with a diameter greater than 3 cm. However, with high insect population densities, the medium–high portions of the trunk can also become infested [3,29]. It is also able to attack seedlings [30]; however, no TCD infestations on nursery stock have been reported [3]. Temperatures higher than about 48 °C and lower than about −18 °C are lethal for the beetle; however, bark may play a role in protecting individuals, as adults and larvae can survive under the bark for short periods at temperatures even lower than −18 °C [31].

## 4. Interactions between *Geosmithia morbida* and *Pityophthours juglandis*

The association between GM and the WTB was established by Newton and Fowler [3] and Tisserat et al. [6] in Colorado in 2008, where GM was consistently isolated from cankers surrounding WTB galleries, as well as from the bark beetle itself. The interactions occurring between the insect and the fungus are not yet well understood. The WTB does not possess mycangia [19]; thus, it carries fungal spores on its elytra [3]; however, the association with GM seems very strong. Based on knowledge from other insect–fungus interactions, various hypotheses have been made: the fungus might be used as a nutritional source by the insect, or it might give other advantages, increasing the success of beetle attacks [32]. In any case, WTB adults exploit not only aggregation pheromones and walnut volatiles to find suitable hosts but also GM volatiles [33], showing how close the relationship between the two organisms is.

## 5. Potential Alternative Vectors of *Geosmithia morbida*

Interestingly, recent studies by Moore et al. [34] and Chahal et al. [35] have revealed that GM has also been isolated from other species of the Curculionidae family, such as *Xylosandrus crassiusculus* (Motschulsky), *Xylosandrus germanus* (Blandford), *Xyleborinus saxesenii* (Ratzeburg), and *Stenomimus pallidus* (Boheman), captured in the vicinity of *J. nigra* trees. These xylophages are known to colonize diseased black walnut trees and could therefore act as alternative vectors for TCD. Furthermore, Moore et al. [34] have also detected GM via molecular methods on other beetle species belonging either to various subfamilies of the Curculionidae family (Cossoninae, Dryophthorinae, Molytinae, and Scolytinae) or to other beetle families, such as Cerambycidae and Bostrichidae. These data suggest the fungus could be transported by a wide range of vectors. However, only the WTB is known to be capable of the natural transmission of the fungus [11], thereby causing TCD, while the potential role of other insect vectors in this disease needs to be further investigated [34,36].

## 6. Symptomatology

Visible symptoms of the disease include leaf yellowing and withering, with flag-like leaves that remain attached to the branch (Figure 2) [15,37]. Subsequently, thinning occurs in the highest portions of the crown, followed by extensive branch mortality. Crown transparency and dieback rapidly progress towards a general decline for the tree. *Pityophthorus juglandis* entrance/emergence holes (1 mm diameter) are visible on the bark of infested branches, in some cases surrounded by a dark amber stain. Internal symptoms include extensive cankering around the beetle galleries [6,8,16,18]. These are often limited to the phloem and the bark, not extending to the inner cambium. After an intense infestation by *P. juglandis*, the cankers merge, ultimately girdling the branches and causing tree mortality (Figure 3). Inside the galleries, it is also possible to observe conidiogenesis, with fungal conidia that are white to light brown in color [15]. According to WTB behavior, the symptoms occur initially on the lower portion of the branches [28], especially those with a diameter greater than 3 cm [38]. Later they spread to the medium-high portions of the trunk [3,29]. In addition, cracks sometimes form on the bark of the small-diameter branches near the tunnels, which gives the branches a rough appearance. Moreover, vigorous epicormic shoots may develop below dead branches, giving the tree a bushy appearance. Though symptom development is slow and progressive, TCD is generally lethal to the black walnut, leading to host death within approximately 8–10 years from initial infection [39].

## 7. Susceptible Hosts

TCD infects hosts in the genera *Juglans* and *Pterocarya* (Juglandaceae) [40,41]. Among the members of the *Juglans* genus, the black walnut is reported in the United States as the most susceptible to TCD, with all the other species showing lower and variable susceptibility degrees [42]. Furthermore, some intraspecific variability in TCD susceptibility has been observed in several walnut species [42]. In addition, there is an uncertainty on the susceptibility level of grafted trees, as it could vary considerably depending on the rootstock [42]. Some North American walnut species, particularly western species (from Arizona and California), are the most tolerant, with only minor cankers on small branches [42]. To date, little is known about the susceptibility of *J. regia*, a species widely planted in Europe for the quality of its wood and for nut production. However, *J. regia* has developed cankers following artificial inoculation with GM in the US, albeit with varying degrees of susceptibility. *Juglans regia* trees naturally infected with TCD were found in field observations both in the US and in Italy [39,42,43]. The problem could further worsen due to the extensive plantation in Europe of many walnut hybrids as ornamental trees or for timber production. Many of these hybrids turned out to be susceptible to TCD in the field, in collections, or following artificial inoculation with *G. morbida* [42,44].

## 8. TCD Integrated Management

### 8.1. Diagnosis

The diagnosis and detection of GM and the WTB are fundamental to preventing the further spread of the disease in walnut plantations and groves in Europe, and they are crucial steps for actuating management measures. The simplest and fastest way to diagnose TCD is via visual examination in the field, although this must be followed up by a more detailed identification of the two members of this insect–fungus complex in the laboratory to avoid species misidentification. This identification can be carried out with conventional methods (macro- and micromorphological analyses), ideally supported by more discriminating molecular approaches. In fact, the fungus grows very slowly in culture and, hence, it is often overgrown by the bacterial and fungal contaminants that commonly coinhabit internal tree tissues and galleries. Furthermore, GM also exhibits pleomorphism that is dependent upon repeated subculturing and physiological growth conditions, and this makes the phenotypic characterization of its colonies difficult [16,41,45]. Moreover, the morphological identification of the beetle can also be troublesome. While bark beetle adults are recognizable through their specific morphological characteristics (by experienced entomologists), immature stages are more difficult to identify, and the complete morphological identification keys for larvae and pupae are lacking [46]. All these hindrances to accurate discernment, coupled with the need for effective and fast diagnostic tools, have made it imperative to develop molecular assays to rapidly detect both members of this insect/fungal complex from infected plant material and from other commodities. In addition, with DNA-based identification being quite sensitive and specific, it should always be preferred for such harmful quarantine organisms as GM and the WTB [47]. 

Various protocols for the diagnosis of TCD have been developed in recent years [34,48,49,50,51,52]. Firstly, Lamarche et al. [48] have developed an efficient and sensitive real-time PCR assay based on TaqMan chemistry that is highly specific, targeting a discriminant part of the GM β-tubulin gene. In addition, this method amplified GM DNA in a concentration as low as three copies of the target gene region; therefore, it is sensitive enough to recognize the pathogen in environmental samples. The same target gene region has also been exploited by Moore et al. [34], who developed a traditional PCR protocol to detect GM DNA from different insect species. Another PCR-based approach, targeting the microsatellite loci previously identified among GM and WTB populations (respectively GS004 and WTB192) [45,53], has been implemented in a protocol for TCD quarantine assessment and management [49]. This PCR procedure is a routine low-cost test and is highly sensitive and specific, but its results may be affected by extraction-phase efficiency. Subsequently, other extraction methods were further investigated by Rizzo et al. [50], who developed a protocol to extract GM and WTB DNA from different matrices with high yields. The DNA was then used by these authors to implement a qPCR probe-based assay that is able to simultaneously detect both organisms while maintaining high sensitivity and reducing the time required for the diagnostic process. A further step forward was taken by Stackhouse et al. [54], who developed an easy-to-use and affordable diagnostic test for both the fungus and its vector based on the visualization of blue-light electrophoresed amplicons and TaqMan probes. In moving forward, the next diagnosis challenge has become the implementation of the point-of-care features of molecular-based methods without losing sensitivity, specificity, and accuracy [55]. Since the loop-mediated isothermal AMPlification (LAMP) reaction offers a wide range of possibilities for point-of-care application, its use is generally preferred as a field-deployable molecular approach [56]. Recently, LAMP-based assays able to detect GM and WTB DNA in a short time (10-15 min) have been developed [51,52]. In particular, these two assays were highly specific, with a sensitivity as low as 3.2 pg/µL and 0.64 pg/µL, respectively, allowing for the rapid, field-deployable screening of environmental samples to identify both the pathogen and the vector (even when only larvae and insect frass are available). These methods, developed by Rizzo et al. [51,52], constitute an opportunity for the rapid, specific, and sensitive diagnosis of GM and the WTB. These protocols, when taken together, constitute a powerful molecular toolbox that could support TCD monitoring and phytosanitary inspections at vulnerable sites (e.g., nurseries, harbors, airports, loading stations, storage facilities, and wood-processing companies). Indeed, their application could help limit the further spread of the disease, preventing new introductions into uncontaminated areas and countries [51,57].

### 8.2. Prevention (Surveillance and Monitoring)

In Europe, GM and its vector (WTB) are Union quarantine pests listed in Annex II part B of the Commission Implementing Regulation (EU) 2019/2072 [58]. A pest risk analysis (PRA) carried out by the European and Mediterranean Plant Protection Organization (EPPO) in 2015 scored the risk of a possible spread of TCD in the European area as “very high” [11]. The current-day pervasiveness of TCD in various areas of central and northern Italy confirms this forecast [13]. Thus, it is essential to implement all the regulatory measures available to prevent TCD entry in uncontaminated areas. 

According to the above-mentioned PRA, the major introduction pathways of TCD are *Juglans* and *Pterocarya* plants for planting and wood, with or without bark [11]. Consequently, the European Union has tried to regulate the importation and movement of *Juglans* commodities within the Union territory by issuing stringent regulations. For example, the Commission Implementing Regulation (EU) [59] includes, in its list of high-risk plants, “plants for planting, other than seeds, in vitro material and naturally or artificially dwarfed woody plants for planting of *Juglans* originating from all third countries”. Furthermore, special requirements for the importation and movement of these commodities within EU territory are listed in Annexes VII and VIII of the Commission Implementing Regulation (EU) 2019/2072 [58]. 

The phytosanitary surveillance of plants for planting is very challenging, especially at the initial stages of TCD infection, when few signs of the beetle (e.g., entry holes) can be accurately identified. Plants for planting seem to have been the main entry pathway of this disease into Italy [11]. This was exacerbated by the import of numerous batches of black walnut seedlings from the US starting in the 1990s, as several black walnut plantations were established with the financial contribution of the EU in various European countries [15]. Currently, black walnut settings are estimated to cover a total area of about 20,000 hectares across 14 countries in Europe [1]. Although black walnut occurs as an ornamental tree in parks, gardens, and avenues, it is mainly cultivated for timber production [1]. The massive importation of black walnut in Europe from the US likely resulted in repeated introductions of the disease, as confirmed by the multiple haplotypes of the beetle found in the TCD invasion areas in Italy [15]. In fact, all the haplotypes in Italy correspond to the same haplotypes in several infested areas in the US [60]. Therefore, phytosanitary inspections of this material at entry points must be conducted with great care by experienced personnel. 

Another commodity that potentially introduces TCD into Union territory is woody plant materials. The likelihood of TCD entry is higher for wood with bark, whereas bark-free wood and squared wood are considered to represent a moderate or low risk [11]. Although posing an even lower risk, other wood materials, such as packaging material, namely packing cases, boxes, crates, drums, pallets, box pallets, and other loading boards, are also regulated. As per Regulation (EU) 2019/2072 [58], all these woody plant materials (if coming from the US) can be imported only with an official statement that they meet one of the following conditions: (a) they originate from areas that are free from GM and its vector: the WTB; (b) they have been squared to entirely remove the natural rounded surface; (c) they have undergone the appropriate heat treatment (and thus marked ‘HT’) to achieve a minimum temperature of 56 °C for a minimum duration of 40 continuous minutes throughout the entire profile of the wood, which ensures the death of the insect [61]. 

However, heat treatment reduces the quality of black walnut logs, making them unsuitable for some wood transformation processes, such as for veneer. Therefore, attention has been focused on alternative treatments for infested logs that kill the two agents of TCD without negatively affecting wood quality. Among the protocols that have been developed in the US for the treatment of infested logs, vacuum steam treatment is one of the most promising, achieving complete elimination of both GM and the WTB by applying a minimum of 56 °C for at least 30 min [62]. Specific surveillance and monitoring techniques consist of visual examinations and pheromone traps in both the US [63] and Europe. These approaches are fundamental in the monitoring programs of insect-infested areas in order to more reliably and accurately detect the presence of TCD and to delimit its distribution. In addition, intensive monitoring of the areas surrounding disease foci should be conducted to prevent the further spread of TCD. The density of the traps depends on the aim; in order to monitor incipient beetle populations, a lower trap density is sufficient; on the contrary, to assess the extent of an established population, the trap density must be higher [39]. 

Trapping is an underutilized approach to prevent new WTB introductions, while it would improve early detection, especially at vulnerable sites, such as points of entry (e.g., ports) and facilities (e.g., mills) receiving *Juglans* wood and plants. In a case study in central Italy, for example, a new outbreak was discovered thanks to pheromone traps, which were followed up by sampling the symptomatic branches to detect beetle entry/exit holes and the associated fungus [15,16]. Pheromone traps were subsequently inspected biweekly for the whole growing season, especially in the period when temperatures exceeded 18–19 °C [44]. Although pheromone traps have proven to be sufficiently effective in monitoring the beetle [64], even at low infestation levels [65], they cannot be employed to control it [11].

### 8.3. Eradication and Containment

TCD is a slow and progressive disease [66], whereby the incidence and severity increase steadily with the growth of the vector population. The long lag time (usually three–five years) between tree infestation and TCD symptom expression allows the disease to establish itself and to spread over the territory before its presence is discovered, and this often nullifies eradication attempts. Eradication can be most effective when the attacked trees or plantations are widely separated (physically) from other susceptible trees. This is confirmed by the Italian situation, wherein, following the first report of the two organisms, several new outbreaks were found near the site of the first finding in northern Italy [25]. Then, in Tuscany (central Italy), once the disease was first reported, it had already spread over an entire plantation (about 1.5 Ha), with a very high beetle infestation rate. The felling of the whole plantation—which, at the time, was Tuscany ’s first TCD outbreak—appeared to have successfully eradicated the disease. However, we have recently (2022) detected the disease in ornamental black walnuts planted in the nearby urban area of Florence [Authors, pers. Comm.]. 

Disease eradication consists of cutting and removing the attacked trees, with extreme attention paid to the proper handling of the TCD-infective/infested cut trees. In fact, attacked cut trees and dead, broken branches on the ground often contain many thousands of walnut twig beetles. Consequently, walnut plant material that is not used must be destroyed on-site by fire (Figure 4), avoiding any possible transfer, while marketable wood material must be properly treated. Specifically, black walnut logs must be sufficiently dried to become noninfective; this dehydration process generally takes two–three years. During this time, the wood must be isolated, usually by stockpiling the wood on a site far from the healthy walnuts or in storage buildings. This process can be sped up by solarization and tarping logs with clear plastic, which concurrently prevents the dispersion of the beetle. Instead, chipping infected wood is quite risky because it does not completely kill all of the beetles such that, in subsequent warm periods, these active beetles will then disperse into uncontaminated areas through the wood chips [67]. 

The same difficulties encountered with eradication also persist with mitigation/containment attempts, although containment is often more realistic than eradication. The elimination of attacked plant residues or the whole tree to reduce inoculum pressure and beetle population density in infested areas has been proven to play a modest role in TCD management, only moderately slowing disease development [67]. Efforts to control TCD should, instead, focus primarily on reducing WTB population density since *G. morbida* is a weak pathogen that requires massive attacks by its insect vector to seriously impact walnut trees [68]. As regards chemical treatments, both bark-sprayed and trunk-injected pesticides have shown only limited effectiveness [67]. Furthermore, if the walnuts grown for nut production are to be treated, the residues of the pesticide (insecticide or fungicide) in its fruit could be a problem, although this is a common practice in the protection of many crops. Another method to reduce WTB populations consists of girdled trap trees in autumn. After these trees have attracted the WTB adults of the last generation, they must be removed and destroyed during wintertime, when the beetles are inactive, before the emergence of the next generation [68]. 

In addition to all the above eradication/containment measures, in order to be applied to the infested area, other compulsory phytosanitary measures should be enacted in buffer zones to prevent the dispersal of GM and the WTB outside of the outbreak area. For example, the ban on the movement of potentially attacked material, i.e., *Juglans* and *Pterocarya* plants for planting and wood products, as well as the constant control of nurseries producing these [11,12].

### 8.4. Public Outreach

The training of plant owners, arborists, nursery personnel, foresters, and other plant industry workers about TCD (speaking, publishing, and exhibiting educational material) could greatly aid in the rapid identification of new disease outbreaks. Since the black walnut is widely exploited as an ornamental tree in urban greenery, educating the general public could remarkably facilitate the early detection of trees with symptoms. Through the recruitment of such volunteer observers, awareness heightening, and public appeals for help in scouting, the accurate and capillary monitoring of the territory can be achieved [57,67].

## 9. Discussion

Thousand cankers disease poses a serious threat to *J. nigra* and *J. regia* across the US and Europe. Since they are two economically and environmentally important walnut species, the alert level towards this disease is high [6,8,9]. In Italy, the only European country where TCD is currently present outside of North America, there is growing concern that it could also massively attack *J. regia*. This species does not seem to have a high susceptibility to the disease, but the situation could vary in cases of high WTB population density [42]. In addition, variation in susceptibility may also depend on the rootstock employed, as *J. regia* is often grafted; for example, ‘Paradox’ (hybrid of *Juglans hindsii* Jeps. ex R.E. Sm. and *J. regia*) and *J. nigra* are among the most commonly used rootstock for this walnut species [42]. *Juglans regia* has a long tradition of cultivation in Italy, particularly in its southern regions. Here, the spread of TCD would, therefore, represent a major problem. In fact, due to the lack of containment measures and the warmer climate of southern Italy, which is more similar to that of the native range of TCD, GM and the WTB could find here optimal conditions for their reproduction and dispersal [16]. These concerns arise also from GM’s thermotolerance since it has remained viable even at temperatures above 40 °C [8]. 

This new phytosanitary problem is complex and multifaceted, involving biological, environmental, and economic issues; consequently, it requires an integrated pest management approach, which will have to combine interdisciplinary scientific collaboration with technical and organizational efforts. The first fundamental step is the development of simple, accurate, and sensitive diagnostic methods to quickly detect GM and the WTB from plant propagation material and timber, which are the main pathways for the local, regional, and international movement of these alien organisms, as well as from other walnut commodities [49,51,52]. The molecular identification of the fungal pathogen and its beetle vector requires specialized equipment and training, so, in some cases, accurate diagnosis in the field is difficult. In order to overcome these problems, an array of molecular tools have recently been developed to improve the diagnostic capabilities of inspection services and enable the detection of the disease at an early stage, even on-site [34,48,49,50,51,52]. 

Research is critical for supporting TCD management and for boosting innovative approaches. Invasions by non-native species normally consist of sets of individuals taken from their native range that are transported to a novel site and are then released. It is essential to investigate the adaptability of these pioneer invaders to their new environment in order to infer their evolutionary and pathogenetic potential [69]. Additionally, more studies are needed to investigate the level of resistance of the germplasm of these two species (including hybrids), which could be ascertained through susceptibility tests of European plant material.

Further research should also address the feasible biological control of the fungus and/or insect through the use of their natural enemies [11]. When dealing with an insect–fungus complex, attention should possibly focus on biocontrol agents that combine insect-killing activity with protection against pathogens (Carroll, 1988) [70]. Some entomopathogenic fungi are known to be lethal to insects and phytopathogenic fungi [71,72]. Among these, *Beauveria bassiana* (Bals.-Criv.) Vuill. (Hypocreales: Cordycipitaceae) and *Metarhizium anisopliae* (Metschn.) Sorokīn (Hypocreales: Clavicipitaceae) are promising candidates, being fungi endowed with an endophytic lifestyle and having revealed an ability to effectively suppress fungal pathogens, soil-inhabiting insects, and Coleoptera in general [70,73,74]. 

Other antagonistic fungi which have demonstrated the ability to interact with both plants and their biotic stressors are the *Trichoderma* species. Selected strains of *Trichoderma* spp. can act in a variety of contexts and pathosystems, triggering systemic and localized plant resistance to diseases [75] and pests [76]. *Trichoderma* antagonists, which are common inhabitants of the soil and often occur in internal tree tissue as endophytes, are reported to act as both biopesticides and biofertilizers [77]. These beneficial microbes are, in fact, able to promote plant growth and development, favoring a greater expansion of the root system, thereby enhancing water and nutrient acquisition and general tolerance to a variety of biotic and abiotic stresses [77]. For instance, *Trichoderma* is capable of controlling insect pests in various ways: through direct parasitism or by the production of compounds with varied antagonistic properties (insecticidal, antifeedant, or repellent) [76]. The frequent occurrence of *Trichoderma* spp. in TCD-associated fungal communities in the US and their known antagonistic behavior suggests that they could represent a promising option in the management of both GM and WTB populations [78]. Another interesting line of research could be the exploitation of intrageneric antagonism [78], with five found *Geosmithia* species with potential antagonism towards GM.

Regardless, the whole walnut phytobiome could play a role in mediating TCD development. According to Onufrak et al. [79], geographical variation in TCD incidence and severity may depend on differences in the walnut phytobiome. In order to assess this hypothesis, these authors characterized the black walnut phytobiome both in its original range (Indiana and Tennessee) and in its introduction areas (Washington). rDNA-ITS sequencing of fungal and bacterial communities from the caulosphere of healthy and diseased trees actually revealed richer and more diverse microbial communities in those walnuts growing in their native ranges compared to those growing in non-native ranges. 

Another key issue in the development of TCD, as well as in its impact assessment, is the possible antagonistic/synergistic role of co-occurring fungal pathogens. TCD-associated fungi could be secondary or opportunistic pathogens that attack TCD-infected walnuts in the later stages of the disease [80] or could be more aggressive pathogens that contribute to the rate of tree decline. For example, *Fusarium solani* (Mart.) Sacc. has often been associated with decaying TCD-infected walnuts [6,80]. This fungus, isolated from the necrotic bark of *J. nigra* and *J. regia* in both Colorado [6] and Northern Italy [80], proved to behave as an early colonizer and a contributing pathogen. 

A field still largely to be explored for the biocontrol of TCD is the exploitation of nematodes. Various species of nematodes were found inside *J. nigra* trees colonized by the WTB in the US. The most frequent species were *Bursaphelenchus juglandis* Ryss, Park., Álv., Nad., and Subb., *Ektaphelenchus* sp., and *Panagrolaimus* sp. [81]. *Bursaphelenchus juglandis* may be a TCD synergist associated with the WTB, while *Ektaphelenchus* spp. are entomoparasites that are associated with various bark beetles. The abundance of these nematodes increases during the later stages of tree decline [81]. Both are typically phoretics and feed on fungi growing in bark beetle galleries [82]. However, their role in TCD development is still unclear [81]. As regards *Panagrolaimus* sp., it was ascertained that it could reduce the cankered area and, thus, TCD severity directly by feeding on GM and indirectly through metabolite production; in addition, it could also limit the fecundity of the WTB [81]. *Panagrolaimus* sp. could be particularly interesting as it is assumed to have the ability to undergo anhydrobiosis, a characteristic that could simplify the use of this species for commercial applications [81].

In conclusion, TCD is the result of a complex, tripartite interaction among the walnut host(s), the fungus GM, and the WTB, for which appropriate management measures have not yet been devised. As we have seen above, current knowledge about this new pest complex is limited and fragmented. Though several control methods (chemical control, biological control, semiochemicals, and the use of resistant cultivars) have been investigated in the US, to date, none of them have proven capabilities for effectively protecting trees from TCD, either preventively or curatively. Hence, controlling TCD is more likely to be successful if all available measures are exploited in an integrated manner.

## Figures and Tables

**Figure 1 pathogens-12-00164-f001:**
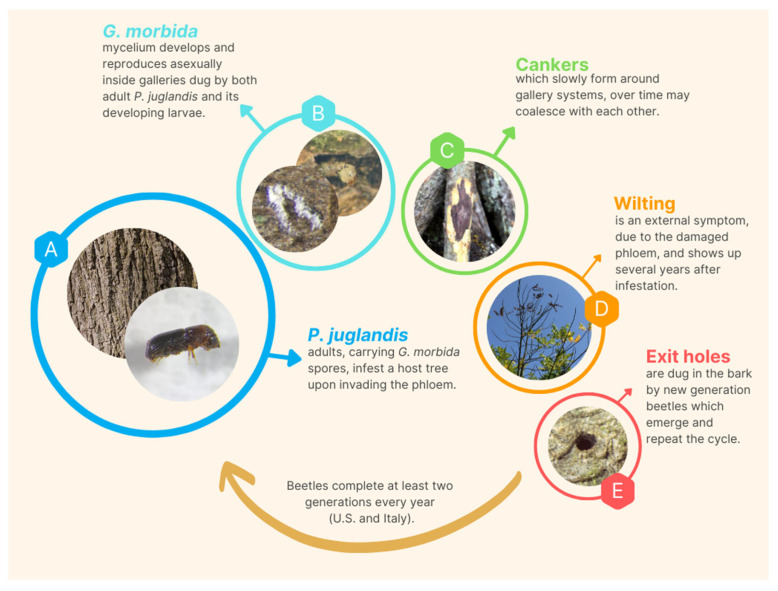
Life cycle of the fungus *G. morbida* and its beetle vector *P. juglandis*, with the most noticeable symptoms on its main host *J. nigra*. The figure outlines the interactions between the two organisms from the first introduction of the fungus into the host through the entry holes until it is led out again by the beetle through its exit holes. *G. morbida* reproduces massively inside the tree as the beetle infestation level increases. At the final stage of colonization, the walnut tree shows a general decline, with clearly visible symptoms.

**Figure 2 pathogens-12-00164-f002:**
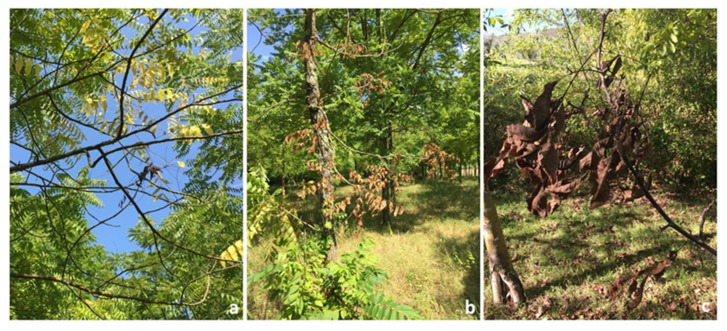
Typical TCD symptoms on black walnut. (**a**) Branches with yellowing symptoms; (**b**) young tree with wilting of lower branches; (**c**) symptoms of flagging on a lateral branch.

**Figure 3 pathogens-12-00164-f003:**
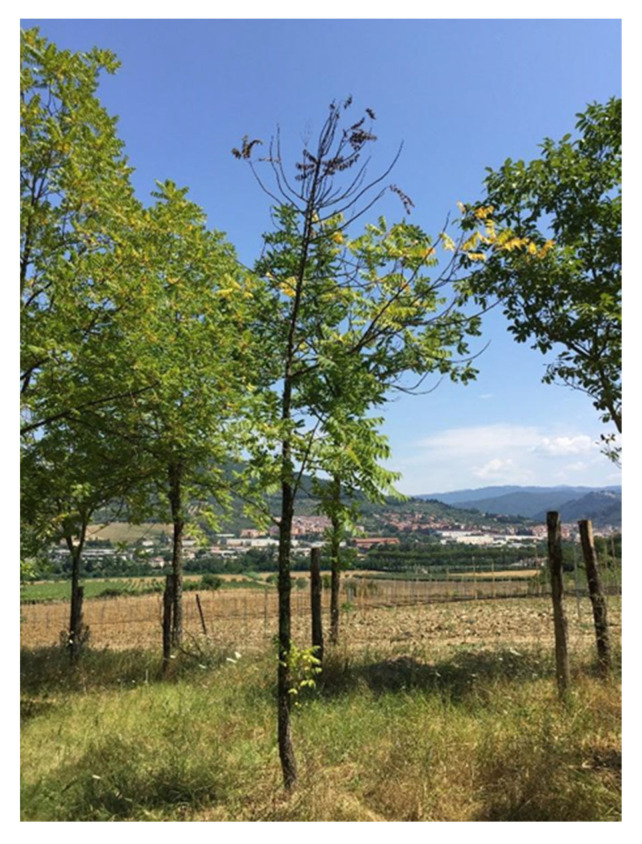
Rapidly wilting *J. nigra* in the final stage of TCD in a plantation in Tuscany (Italy).

**Figure 4 pathogens-12-00164-f004:**
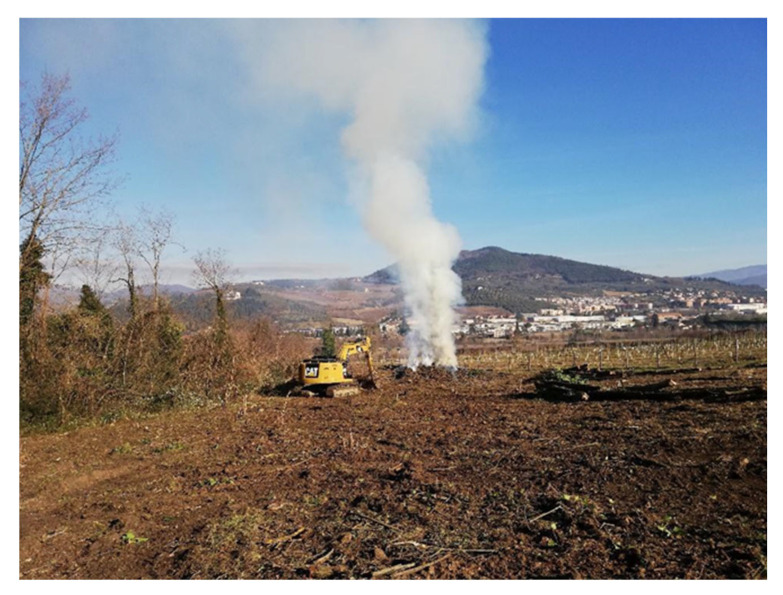
TCD eradication intervention in Tuscany (Italy).

**Table 1 pathogens-12-00164-t001:** Chronology and location of GM and WTB reports in Italy.

Year of Records	Region	Province	Organism Found	Attacked Host	Stand Type *	Mode of Detection **	Symptom Occurrence ***
2013–2020	Veneto	Vicenza Treviso Venezia Padova Rovigo Verona	WTB and GM	*J. nigra* *J. regia*	Single tree Tree grove Plantation	V + T	Yes
2015–2016	Piedmont	Turin Novara Cuneo Vercelli	WTB and GM	*J. nigra*	Single tree Tree row Plantation	V + T	Yes
2015	Friuli Venezia Giulia	Pordenone	WTB	-	-	T	Not
2015–2016	Lombardy	Not indicated	WTB and GM	Not indicated	Single tree	V + T	Yes
2018–2022	Tuscany	Florence	WTB and GM	*J. nigra*	Single tree Plantation	V + T	Yes
2019	Emilia Romagna	Reggio Emilia	WTB and GM	*J. nigra*	Tree row	V	Yes

* type of stand where the organisms were found; ** mode of detection: visual (V) = if attacked trees were found; trap (T) = if only beetle adults were captured in traps; *** yes = symptomatic trees; not = no symptomatic trees.

## Data Availability

No new data were created or analyzed in this study. Data sharing is not applicable to this article.

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
