# Peer review of "Thousand Cankers Disease in Walnut Trees in Europe: Current Status and Management"

_pathogens, 2023, doi:10.3390/pathogens12020164_

Round 1
Reviewer 1 Report
Ln12 Italicize Juglans nigra for consistency
Ln42 Drop “a couple of years later, in 2013”, REVISE “Thousand cankers disease was first reported in Europe specifically in northern Italy (Veneto region) in 2013.”
Ln51 Table and table description do not match; drop NOT and make it NO in TABLE or change to NO in table description.
FIG 1 Description for G. morbida should read MYCELIUM, not micellium.
Ln63 Citation needed? “although it is reported”
Ln92 Unclear which portions of [16-19] support this sentence 91-92? Each of these papers takes a different approach in defining larvae.
Ln106 drop “for the time being” Consider. “However, only PJ is known to efficiently transmit…”
Ln108 order [6, 22, 21] out of sequence consider [6, 21, 22]?
Ln130 lumping citations to support time to death is confusing to reviewer, consider the most current and dropping some of these references.
Ln135 Italicize Juglans for consistency
Ln161 consider lacking or non-existent over inexistent?
Ln163 drop able and replace early with “rapidly”
Ln167 again for this reviewer something is lost by lumping [34-38] the various protocols; consider revising with specific findings from each reference; compare contrast; etc…
Ln193 Rizzo et al. [37] add the reference here.
Ln217 do we retain just disease? or is this disease/complex throughout manuscript?
Ln218 drop “may show up” add “can be accurately identified”
Ln275 “sped” up; drop speeded
In 7.3 you may wish to consider a discussion on “steam-vacuum” approaches highlighted in https://academic.oup.com/jee/article/114/1/100/6020108
Ln295 delete space between destroyed during
Ln322 change comma to . after U.S.,
Ln353 delete 353 to beginning of 356 “As with all recently introduced diseases which, being new, are little known, so too for this disease complex, research is what fills the gaps; in our case research must fill in the knowledge gaps about GM and PJ biology and ecology.”
Ln363 Consider making the final concluding statements start as new paragraph Ln364 to improve reading and closure.
Overall Impressions:
-This paper reviews current status of TCD management; perhaps a bit more in the introduction along the lines of why this review is needed or what will be gained by doing this review
-This reviewer had some issues with lumping of references, while it pads the reference section each reference should be daylighted to its specific relevance and context in the line.
-A fair amount of work has been done with steam vacuum approaches and should be added in section 7.3
https://academic.oup.com/jee/article/114/1/100/6020108
Author Response
Ln12 Italicize Juglans nigra for consistency
Done
Ln42 Drop “a couple of years later, in 2013”, REVISE “Thousand cankers disease was first reported in Europe specifically in northern Italy (Veneto region) in 2013.”
Done
Ln51 Table and table description do not match; drop NOT and make it NO in TABLE or change to NO in table description.
Done
FIG 1 Description for G. morbida should read MYCELIUM, not micellium.
Done
Ln63 Citation needed? “although it is reported”
Done
Ln92 Unclear which portions of [16-19] support this sentence 91-92? Each of these papers takes a different approach in defining larvae.
Done.
Ln106 drop “for the time being” Consider. “However, only PJ is known to efficiently transmit…”
Done
Ln108 order [6, 22, 21] out of sequence consider [6, 21, 22]?
Done
Ln130 lumping citations to support time to death is confusing to reviewer, consider the most current and dropping some of these references.
Done. We have removed superfluous quotes and, as you suggested, we have left only the most recent one
Ln135 Italicize Juglans for consistency
Done.
Ln161 consider lacking or non-existent over inexistent?
Done.
Ln163 drop able and replace early with “rapidly”
Done.
Ln167 again for this reviewer something is lost by lumping [34-38] the various protocols; consider revising with specific findings from each reference; compare contrast; etc…
These quotes are lumped together because this is the opening sentence of the paragraph. Each individual publication is cited and discussed below, within the paragraph.
Ln193 Rizzo et al. [37] add the reference here.
Done.
Ln217 do we retain just disease? or is this disease/complex throughout manuscript?
Done.
Ln218 drop “may show up” add “can be accurately identified”
Done.
Ln275 “sped” up; drop speeded
Done.
In 7.3 you may wish to consider a discussion on “steam-vacuum” approaches highlighted in https://academic.oup.com/jee/article/114/1/100/6020108
Done. See paragraph inserted in 8.2. Prevention (surveillance and monitoring), lines 501-507
Ln295 delete space between destroyed during
Done.
Ln322 change comma to . after U.S.,
Done.
Ln353 delete 353 to beginning of 356 “As with all recently introduced diseases which, being new, are little known, so too for this disease complex, research is what fills the gaps; in our case research must fill in the knowledge gaps about GM and PJ biology and ecology.”
Done.
Ln363 Consider making the final concluding statements start as new paragraph Ln364 to improve reading and closure.
Done.
Overall Impressions:
-This paper reviews current status of TCD management; perhaps a bit more in the introduction along the lines of why this review is needed or what will be gained by doing this review
Done.
-This reviewer had some issues with lumping of references, while it pads the reference section each reference should be daylighted to its specific relevance and context in the line.
Done.
-A fair amount of work has been done with steam vacuum approaches and should be added in section 7.3
Done. See above. Paragraph inserted in 8.2. Prevention (surveillance and monitoring), lines 501-507
Reviewer 2 Report
The manuscript represents a summary of the current understanding of the Thousand Cankers Disease of Walnut trees in Europe. As a review paper, does not provide new data; however, these type of papers are useful to obtain a tighter and historical updated information for readers interested in the disease. From this point of view, the paper has a practical value especially, since the EPPO rate the risk of a possible spread of the disease in Europe as very high.
However, reading the manuscript I have the impression that there is no an in-depth review of the literature (there are 126 papers of the disease in WOS, and only 38 dealing with TCD were cited here). It is true that the review focuses on prevention and management, but I do not that the contribution of the article goes much beyond to what was already indicated by EPPO.
Title: Thousand cankers disease of Walnut trees in Europe: status management and challenges
I do not really see much about challenges or needs of future research
Abstract:
Based on the data provide in Table 1, it may be that the first sentence “… which is plaguing commercial plantations…” overestimates the current status of the disease in Europe and may also overestimates the actual damage of the disease, when later the fungus is described as an apparently harmless wound pathogen (Line 72)...or a weak pathogen that requires massive attacks by its insects vector to seriously impact walnut trees (Line 287).
The abstract also states “… after an analysis of the main biological and ecological traits of both members of the insect/fungus complex”, however I have the impression that they focus more on the anatomical description of the insect and the fungus than in their biology or ecology.
Introduction
Consider headings 2-6, as possible subheadings of the Introduction.
I do not see Table 1 cited in the text. I would be of great help to have a map that represents the distribution of the disease in Italy or provide a value of the area attacked by the complex in the different plantations.
Legend of Figure 1. Replace “beetle accomplice” by “beetle vector”
The authors use different abbreviations for the insect name throughout the manuscript: PJ or WTB, please use the same term, preferably WTB as it match the common name “Walnut Twig Beetle”.
Lines 75 and 76. there are also cases where both the insect and the fungus were detected separately, this should also be included in the review
Line 79-92. Here some information of the biology and ecology of the insect seems pertinent, climatic preferences, trophic preferences, how it vector the fungi?, has a mycangium? How strong is the aggregation pheromone? Prefer healthy trees, weak trees? etc.
Line 106: ...”is the only insect attacking healthy trees” needs a specific references
Line 107: references are not in order
Lines 134-143. Attacks also walnut hybrids that are being extensively planted in some areas of Europe?
Lines 146-198. The authors seem to highlight the detection method of Rizzo et al. in which some are also co-authors. If so, the reason why this method is better than others should be clearly explained to avoid bias.
Line 219: “numerous batches” “several walnut plantations” can you be more precise?
Is the insect common on seedlings? If so, it should also be included under its biology and preferences in the “The vector Pityophthorus juglandis” section.
Line 239. I understand that the aggregation pheromone of this bark beetle is not very strong. More information is needed on how this may influence the success of pheromone trap monitoring techniques.
Discussion: The first paragraphs of the discussions seem more appropriate for the introduction when the authors raise the problem and its significance.
I think more information is needed about the “challenges” stated in the title.
Author Response
However, reading the manuscript I have the impression that there is no an in-depth review of the literature (there are 126 papers of the disease in WOS, and only 38 dealing with TCD were cited here). It is true that the review focuses on prevention and management, but I do not that the contribution of the article goes much beyond to what was already indicated by EPPO.
Literature was enriched with new quotes relating to the new topics that were added and discussed.
Title: Thousand cankers disease of Walnut trees in Europe: status management and challenges
I do not really see much about challenges or needs of future research
See our latest answer to your questions (at the bottom)
Abstract:
Based on the data provide in Table 1, it may be that the first sentence “… which is plaguing commercial plantations…” overestimates the current status of the disease in Europe and may also overestimates the actual damage of the disease, when later the fungus is described as an apparently harmless wound pathogen (Line 72)...or a weak pathogen that requires massive attacks by its insects vector to seriously impact walnut trees (Line 287).
It is true that the fungus is a weak pathogen but if you consider it alone. The danger comes from the lethal association with the beetle. In fact, the two organisms together cause enormous damage in the US, as we have specified (see, for example: Griffin, G. J. 2015. Status of thousand cankers disease on eastern black walnut in the eastern United States at two locations over 3 years. Forest Pathol. 45: 203–214; 27; Seybold, S. J.; Klingeman, W. E.; Hishinuma, S. M.; Coleman, T. W.; Graves, A. D. Status and impact of walnut twig beetle in urban forest, orchard, and native forest ecosystems. Journal of Forestry, 2019, 117(2), 152-163)
The abstract also states “… after an analysis of the main biological and ecological traits of both members of the insect/fungus complex”, however I have the impression that they focus more on the anatomical description of the insect and the fungus than in their biology or ecology.
Done. We have integrated both paragraphs on the fungus and the beetle and have included important information on their biology and ecology. We also created a new paragraph: "Interactions between Geosmithia morbida and Pityophthours juglandis"
Introduction
Consider headings 2-6, as possible subheadings of the Introduction.
We do not agree with your suggestion to place headings 2-6, as subheadings of the Introduction. This is because the various headings, with their titles, immediately bring the reader to the various topics covered
I do not see Table 1 cited in the text. I would be of great help to have a map that represents the distribution of the disease in Italy or provide a value of the area attacked by the complex in the different plantations.
Done.
Legend of Figure 1. Replace “beetle accomplice” by “beetle vector”
Done.
The authors use different abbreviations for the insect name throughout the manuscript: PJ or WTB, please use the same term, preferably WTB as it match the common name “Walnut Twig Beetle”.
Done.
Lines 75 and 76. there are also cases where both the insect and the fungus were detected separately, this should also be included in the review.
We have reported cases of G. morbida isolation from other insect species in the section “Potential alternative vectors of Geosmithia morbida”. As regards detection of WTB separately from GM we found no reference. To our knowledge the insect has been found alone only in traps, but not on attacked trees. If you Know articles reporting this, please give us this reference. We will be happy to discuss this topic in our review.
Line 79-92. Here some information of the biology and ecology of the insect seems pertinent, climatic preferences, trophic preferences, how it vector the fungi?, has a mycangium? How strong is the aggregation pheromone? Prefer healthy trees, weak trees? etc.
Done. This information has been reported.
Line 106: ...”is the only insect attacking healthy trees” needs a specific references
We had not written that P. juglandis "is the only insect attacking healthy trees". We had written that P. juglandis is the only insect known to efficiently carry the fungus. Now we have reformulated the sentence to try to make this concept clearer.
Line 107: references are not in order
Done
Lines 134-143. Attacks also walnut hybrids that are being extensively planted in some areas of Europe?
Done. We added a sentence where we raised the question of hybrids.
Lines 146-198. The authors seem to highlight the detection method of Rizzo et al. in which some are also co-authors. If so, the reason why this method is better than others should be clearly explained to avoid bias.
The reason why the work of Rizzo et al. seems to be more performing is indicated in the text: "simultaneously detect both organisms, while maintaining high sensitivity and while reducing the time required for the diagnostic process". However, since we are among the authors, we have toned down, removing some adjectives (efficient and reliable).
Line 219: “numerous batches” “several walnut plantations” can you be more precise?
It is quite difficult to accurately quantify the number of batches and plantations because the importation of black walnut from North America started at the beginning of the seventeenth century and has had a strong surge since the ‘90s of the last century through EU funding. However, we have reported the numbers/hectares that were estimated in a recent review.
Is the insect common on seedlings? If so, it should also be included under its biology and preferences in the “The vector Pityophthorus juglandis” section.
done
Line 239. I understand that the aggregation pheromone of this bark beetle is not very strong. More information is needed on how this may influence the success of pheromone trap monitoring techniques.
Done. I added a sentence on this regard. See lines 523-524
Discussion: The first paragraphs of the discussions seem more appropriate for the introduction when the authors raise the problem and its significance.
Done.
I think more information is needed about the “challenges” stated in the title.
By "challenges" we mean efforts to contain disease; at a regulatory/technical level (e.g. surveillance and monitoring), on the one hand; and research efforts, especially the development of robust diagnostic methods, on the other. However, we took the "challenges" out of the title (because "management" encompasses everything).

Reviewer 3 Report
Dear Authors,
I went through your paper and it seems like a nice collection of data on TCD. However, what is lacking is, you have not discussed well about the current research, all about is giving facts rather than discussing potentials compared to other fungal-insect combination taking and examples as well. In addition, Specific comments are given in the revised files.

Author Response
I do not agree with the way you have presented the discussion. At the begin of this section, is'a again all facts rather than you are talking what are the problems associated with each, why you think these methods need to improve, what are suggestions.
Since this reviewer made his comments and suggestions, directly on the manuscript, we have duly amended it according to all his requests and suggestions. In addition, we lengthened the discussion and integrated it by adding new issues

Round 2
Reviewer 2 Report
The manuscript is much improved compared to the first version; the authors did a great job collecting information that makes the review more complete. I still miss some information about the impact of the disease in Italy in terms of number of hectares or trees attacked, but it may not be easy to know. There are still some minor corrections, such as italicized Latin names on line 467. But overall, I think the manuscript can be accepted now.
Author Response
Regarding the additional data on the size of infestations in Italy we cannot comply to the request by Referee2 since such data are not available in the literature. The remaining minor revisions were taken care of and they are now added to the manuscript.